# Computational Assessment of Cannflavin A as a TAK1 Inhibitor: Implication as a Potential Therapeutic Target for Anti-Inflammation

Sarunya Chuanphongpanich [1], Satapat Racha [1], Banthita Saengsitthisak [2], Pichai Pirakitikulr [3] and Kannika Racha [4,*]

1. Department of Pharmaceutical Chemistry and Natural Medicine, Faculty of Pharmacy, Payap University, Chiang Mai 50000, Thailand; omsarunya@gmail.com (S.C.); satapatto@gmail.com (S.R.)
2. Department of Pharmaceutical Technology and Biotechnology, Faculty of Pharmacy, Payap University, Chiang Mai 50000, Thailand; banthitha.saeng@gmail.com
3. Material Science Research Center, Faculty of Science, Chiang Mai University, Chiang Mai 50200, Thailand; pichaijiang@yahoo.com
4. Department of Pharmaceutical Care, Faculty of Pharmacy, Payap University, Chiang Mai 50000, Thailand
* Correspondence: kannika_i@payap.ac.th; Tel.: +66-53-851-478

**Abstract:** TAK1 (transforming growth factor-beta-activated kinase 1) is a crucial therapeutic target in inflammation-related diseases. This study investigated the inhibitory potential of cannflavin A, a flavonoid found in *Cannabis sativa*, against TAK1. Through in silico approaches, including drug-likeness analysis, ADMET assessment, molecular docking, and molecular dynamics simulation, the binding affinity and stability of cannflavin A were evaluated. The results demonstrate that cannflavin A exhibits excellent ADMET properties and displays superior binding affinity and stability at the ATP binding site of TAK1 when compared to the known inhibitor takinib. Notably, the decomposition of binding free energy unveils critical amino acid residues involved in TAK1 binding, underscoring the inhibitory effect of cannflavin A through TAK1 inhibition. These findings highlight the potential of cannflavin A as a TAK1 inhibitor and its significant implications for the development of targeted therapies in inflammation-related diseases. Through modulating inflammatory signaling pathways, cannflavin A holds promise for more effective and tailored treatment strategies, particularly in rheumatoid arthritis. This study contributes to the current understanding of cannflavin A's application and provides a foundation for further research and innovative approaches in targeted therapies for inflammatory conditions.

**Keywords:** TAK1; cannflavin A; rheumatoid arthritis; inflammation; molecular docking; dynamic simulation



## 1. Introduction

Rheumatoid arthritis (RA) is one of the most chronic systemic autoimmune disorders characterized by persistent inflammation that primarily affects diarthrosis joints. It leads to swelling, pain, remodeling, and immobility, significantly impacting quality of life. Global epidemiological data on RA reveal a worldwide prevalence of approximately 0.1–2.0% [1], with women and elderly individuals more susceptible to the disorder. Prominent evidence shows that tumor necrosis factor (TNF) plays a critical role in regulating pro-inflammatory signaling in the affected joints, synovial fluid, and the serum of RA patients [2]. Immunomodulating agents, such as TNF blocking agents, have been shown to slow down joint damage progression and alleviate some RA symptoms through inactivating the biological function of the TNF signaling pathway [3]. Currently, five TNF blocking agents are approved for managing moderately to severely active RA and other autoimmune diseases [4,5]. The first-generation agents include adalimumab, etanercept,

and infliximab, while the second-generation agents include certolizumab pegol and golimumab [6]. However, up to 40% of all patients do not respond to TNF blocking agents, often due to long-term disease mitigation and immune sensitization to the therapeutic agent [7]. Since TNF blocking agents are ineffective when administered orally and require intravenous administration, noncompliance becomes a significant issue [8]. Moreover, these agents cannot cross the blood–brain barrier (BBB) due to their high molecular weight, rendering them ineffective in treating central nervous system (CNS) diseases associated with TNF signaling, including neurodegenerative conditions [9,10]. Furthermore, TNF blocking agents are associated with an increased risk of bacterial and viral infections [11].

Identifying essential signaling kinases, including transforming growth factor beta-activated kinase 1 (TAK1), represents a novel approach to inhibiting TNF-induced inflammation using a small molecule inhibitor that can be orally administered [8]. TAK1 is essential to the TNF signaling pathway [8,12,13]. TAK1 is activated by TNF binding to TNFR1, leading to phosphorylation and activation of downstream inflammatory pathways involving NF-κβ, p38, and c-Jun N-terminal kinase (JNK), which are associated with inflammation-related conditions [12,13]. Studies using genetic knockout of TAK1 have demonstrated that in the absence of TAK1, it prevents the induction of TNF pro-inflammatory responses in mouse LPS challenge models [14]. Consequently, TAK1 has been identified as a significant regulator of TNF signaling in immune cells. It is a promising therapeutic target for treating inflammatory-mediated conditions such as rheumatoid arthritis [8], inflammatory bowel syndrome [15], renal disease [16], skin inflammation [17], and neuropathic pain [18]. Recent research has shown that inhibiting TAK1 with the TAK1 inhibitor takinib significantly suppresses TNF secretion and immune cell signaling, providing a potential approach to reduce the effects of TNF and alleviate RA symptoms and damage [8]. Through analyzing the crystal structure of human TAK1 in the presence of takinib, an ATP-competitive inhibitor, researchers gain valuable insights into the specific binding interactions occurring at the active site. It is worth highlighting that takinib forms multiple hydrogen bonds with critical residues like LYS63, ALA107, and ASP175, while also engaging in hydrophobic interactions involving VAL42, THR106, and GLY110. These extensive intermolecular associations provide a compelling explanation for the exceptional potency and selectivity of takinib towards TAK1, particularly when compared to closely related kinases [13]. Small molecules that selectively target TAK1 may offer a safer and more feasible option for treating chronic inflammation [13,14]. Therefore, developing small molecules that suppress the TNF signaling pathway through selectively targeting TAK1, administered orally, holds promise for improving RA in clinical outcomes.

Recent research has suggested that flavonoids, a class of phytochemical compounds found in plants, may have antioxidant properties, modulate the gut–joint axis, regulate the immune system, and inhibit inflammatory responses. These properties could be beneficial in treating RA [19]. Cannflavins are prenylated flavonoid compounds found in *Cannabis sativa* L. (Canabaceae family) [20]. Various pharmacological activities of cannflavins have shown promising antioxidant [21], anti-cancer [22], neuroprotective [23], antileishmanial activity [24], anti-viral [25], and anti-inflammatory properties [26–28] in pre-clinical evaluations. To date, three cannflavins from *Cannabis sativa* L. have been identified, including cannflavin A, B, and C [29]. Cannflavin A and B have exhibited potent anti-inflammatory activity through inhibiting prostaglandin E2 (PGE2) [26,28], microsomal prostaglandin E synthase-1 (mPGES-1), and 5-lipoxygenase (5-LO) [27]. Moreover, cannflavin A and B have been reported to have anti-inflammatory benefits that were approximately thirty times more effective than aspirin [26]. However, the effect of cannflavins on TAK1 and its underlying molecular mechanism of action have not been demonstrated. Therefore, we aim to investigate the molecular complexation between cannflavins and TAK1 using ADMET screening, analyze their drug-likeness properties, and perform molecular docking, MD simulations, and free energy analysis to understand the protein–ligand interactions together with drug-likeness and toxicity to identify drug-like compounds. This study is expected to provide helpful information for the application potential of cannflavins as a therapeutic

means for the treatment of RA and support further development of cannflavin-related compounds along with the design of novel drugs.

## 2. Materials and Methods

### 2.1. Drug-Likeness and ADMET Prediction

This study utilized the SwissADME web server to predict drug-likeness according to Lipinski's rule and to determine pharmacokinetic properties, such as absorption, distribution, metabolism, and excretion (ADME) [30]. Furthermore, the ADVERpred web server [31] and ProTox-II web server [32] were employed to predict the toxicity of cannflavins derived from *Cannabis sativa*, including cannflavin A, B, and C. These cannflavins were specifically chosen based on compounds found in the previous literature [29]. Additionally, the cannflavins were examined for any pan-assay interference compound (PAINS) liabilities. All compounds were submitted in canonical SMILES format, obtained from the PubChem database [33], to facilitate the analysis.

### 2.2. Protein Preparation

The crystal structure of takinib in complex with human TAK1 was obtained from the RCSB Protein Data Bank (RCSB.org (accessed on 1 February 2023)) [34] using PDB ID 5V5N [13]. To prepare for the docking simulation, we employed UCSF Chimera 1.15 [35] to assign missing hydrogens, remove water molecules, and separate the protein–ligand complex and added the missing residues using the Modeller 10.3 program [36] with the standard method.

### 2.3. Ligand Preparation

Takinib, a TAK1 inhibitor, was employed as a reference compound in the docking method (PDB ID 5V5N). The cannflavin compounds were obtained in a 3D conformer from the PubChem database and then optimized using the B3LYP/6-31G (d, p) basis set implemented in the Gaussian 09 program [37].

### 2.4. Molecular Docking

AutoDock Vina 1.2.3 [38,39] was used to investigate the molecular interaction between the ligands and proteins. The exhaustiveness parameter was increased to 32, and a grid box of 20, 20, 20 Å in size was used [40]. The binding site was determined using the center of the reference compound obtained from the PDB. Finally, the screened compounds that have better free energies of binding than the reference compound were submitted to MD simulation analysis.

### 2.5. Molecular Dynamics Simulations and Free Energy Calculations

Molecular dynamics (MD) simulation was performed using the Amber18 packages [41] on GPU RTX 2080 Ti according to the previously described method [42]. The FF14SB force field [43] was employed for the protein parameter, while the ligand parameter utilized the Generalized AMBER Force Field version 2 (GAFF2) [44] in conjunction with RESP charge [45]. To ensure proper solvation, all complex systems were immersed in a truncated octahedral box containing pre-equilibrated TIP3P water molecules with a radius of 10 Å. Additionally, in order to maintain system neutrality, three sodium counter ions were added to neutralize the system charges. The systems underwent energy minimization using the initial 2500 steps of the steepest descent (SD) algorithm, followed by an additional 2500 steps employing conjugate gradient techniques with weak restraints. Subsequently, NVT equilibration was achieved at a temperature of 310 K. To maintain the integrity of the hydrogen bonds, the SHAKE method was employed to constrain the movement of the hydrogen bond atoms. Long-range electrostatic interactions under periodic boundary conditions were treated using the particle mesh Ewald (PME) algorithm. Finally, unrestrained molecular dynamics (MD) simulations were conducted for 200 ns using the pmemd.cuda module.

The obtained results were evaluated using various analytical measures, namely root mean square deviation (RMSD), radius of gyration (Rg), solvent accessible surface area (SASA), root mean square fluctuation (RMSF), hydrogen bond analysis, and Gibbs free energy landscape analysis (FEL). These evaluations were performed utilizing CPPTRAJ software. Additionally, the binding free energy for the complex systems was computed by means of employing the molecular mechanics/generalized Born surface area (MM/GBSA) method [46] with the MMPBSA.py module in AMBER18. Furthermore, the visualization of 3D binding interactions was accomplished utilizing UCSF ChimeraX [47].

## 3. Results and Discussion

### 3.1. Drug-Likeness and ADMET Prediction

The evaluation of drug-likeness and ADMET properties plays a vital role in the drug discovery and development process. This stage aims to maximize the drug's effectiveness, safety, and pharmacokinetics. The ultimate objective is to ensure that the drug can reach the intended site in adequate concentrations, facilitating the desired physiological effect while maintaining safety. According to estimates, approximately 40% of drug candidates are eliminated during preclinical tests due to concerns related to ADMET properties [48]. This predicted drug-likeness was assisted through the utilization of SwissADME. Lipinski's rule governs the optimal medication for oral administration. All compounds were found to be in compliance with the prior rules with no violations. According to SwissADME, it did not highlight any PAINS related to the investigated molecules.

The pharmacokinetic properties of cannflavin A, B, and C and the inhibitor takinib were analyzed and are represented in Figure 1A using the BOILED-Egg model to predict absorption, distribution, and excretion [49]. For absorption, all of the screened compounds lie inside the white ellipse with the probability of being absorbed by the human gastrointestinal tract; this also agrees with Lipinski's rule. Cannabis use has been reported to induce long-lasting psychotic disorders, and a dose–response relationship has been observed [50]. Despite having the CNS side effects of cannabis, the capacity of the investigated molecules on the permeability of the blood–brain barrier (BBB) was evaluated. For distribution, all of the screened compounds are predicted to be outside the yellow region (yolk) with considerable amounts not passing to the BBB, which suggests that they have a low potential for causing CNS side effects. Moreover, for excretion, all of the screened compounds are predicted to be non-substrates (red dots) of the permeability glycoprotein (P-gp), which is the most important active efflux mechanism involved in those biological membranes [51,52]. Also essential is knowledge about the interaction of molecules with the microsomal cytochrome P450 (CYP) monooxygenase system [53]. This large family of isoenzymes is a key player in drug metabolism. Inhibition of CYP enzymes is certainly one major cause of pharmacokinetics-related drug interactions, leading to increased drug toxicity or other unwanted adverse effects due to a reduction in metabolism and the augmentation of plasma drug levels from drugs with a narrow therapeutic index [54]. Cannflavin A and C had negligible values, indicating that they are non-inhibitors of CYP450, but cannflavin B is an inhibitor of CYP2C9 and takinib is an inhibitor of CYP2D6 (Figure 1B), with it inhibiting the metabolism of drug substrates.

In silico toxicity prediction intends to supplement the current in vitro toxicity methods through forecasting the harmful effects of chemicals, thus minimizing the duration, costs, and dependence on animal testing for toxicity assessment [32]. The ADVERpred web server was used for the estimation of drug-induced arrhythmia, cardiac failure, myocardial infarction, hepatotoxicity, and nephrotoxicity since they are the most frequent and severe adverse effects, which often lead to death and require the withdrawal of the drug from the market. Figure 1C shows that takinib caused arrhythmia and heart failure. However, the probability of the active (Pa) value of takinib having adverse effects was found to be <0.5, which indicated lower probability of it causing adverse effects. Interestingly, cannflavin A, B, and C did not show any adverse effects. Chronic safety testing in animals, including carcinogenicity and mutagenicity studies, are usually performed concurrently with pre-

clinical tests. The ProTox-II web server was used for predicting endpoint carcinogenicity and mutagenicity, which aid in the development of safe therapeutic agents. Figure 1C shows that the cannflavins had non-carcinogenicity and mutagenicity, except for takinib. Considering all these parameters and data, it is predicted that cannflavin A and C possess favorable drug-likeness and conform to ADMET profiles.

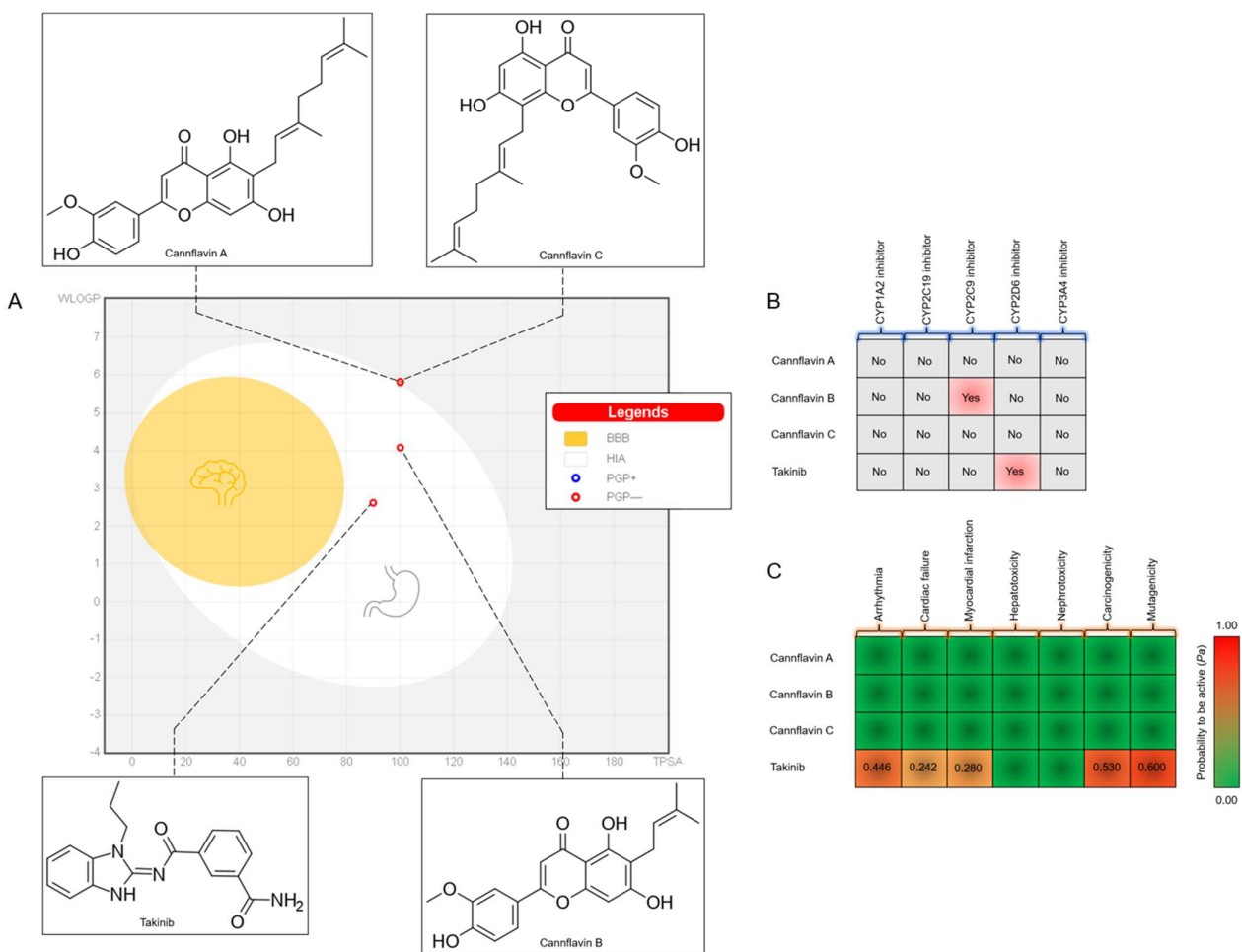

**Figure 1.** ADMET prediction of the screened compounds and the reference control takinib: (**A**) BOILED-Egg model, (**B**) CYP inhibitor profiles, and (**C**) toxicity heat map representation.

### 3.2. Molecular Docking

Molecular docking studies were performed to estimate the interaction pattern and free binding energy of the investigated compounds at the ATP binding site of TAK1. Initially, the native co-crystal ligand (takinib) of TAK1 (PDB ID 5V5N) was redocked to legitimize the virtual molecular docking protocol for its accuracy. The root mean square deviation (RMSD) value between the co-crystal and redocked takinib pose is 1.094 Å, which is considered a successful docking protocol (Figure 2A), and the free energies of binding were measured as −8.742 kcal/mol. A total of three major cannflavins from *Cannabis sativa* were docked to TAK1. The receptor–ligand interactions indicated that all of the compounds used for molecular docking have a substantial binding affinity towards the receptor as compared to the reference compound. The detailed list is depicted in Figure 2B. The compounds with free energies of binding greater than takinib (−8.742 kcal/mol) were taken for further analysis. As presented in the visualization of the compounds' binding poses (Figure 2C), as expected, the compounds are well-contained within the ATP binding site of TAK1. Based on the molecular docking result analysis, the best two molecules (cannflavin A and C) were selected for further molecular dynamics simulation study.

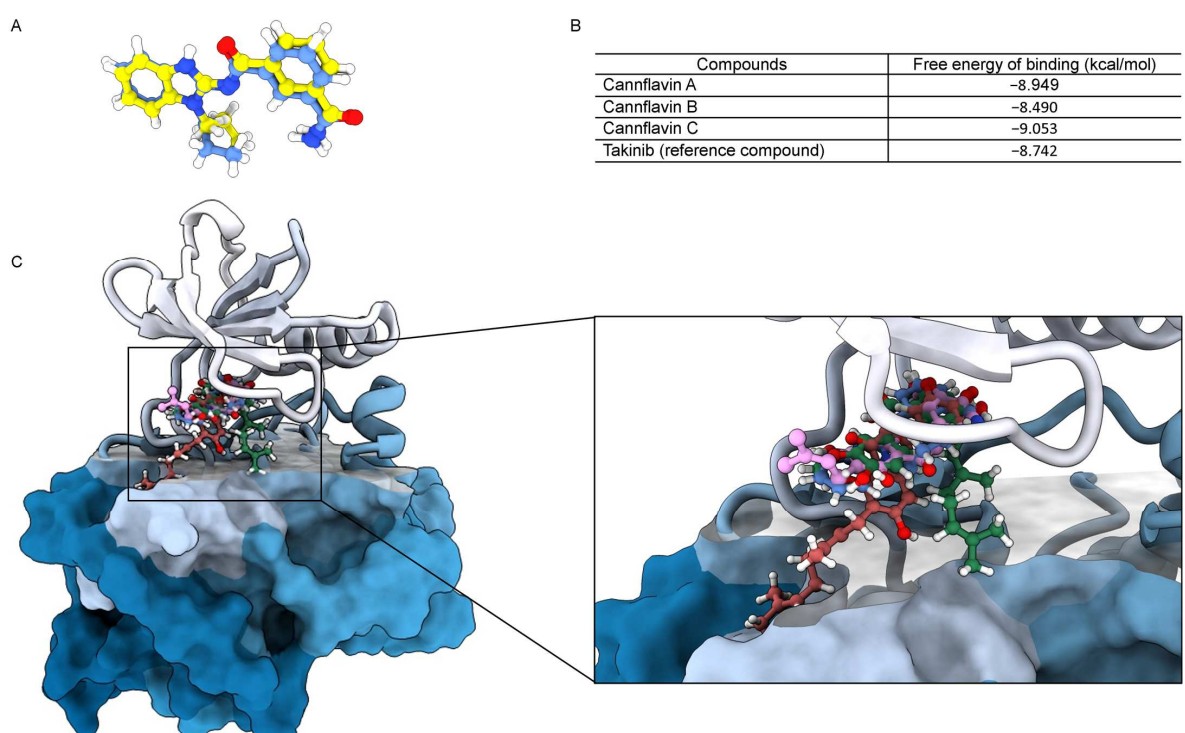

A

B

| Compounds | Free energy of binding (kcal/mol) |
|---|---|
| Cannflavin A | −8.949 |
| Cannflavin B | −8.490 |
| Cannflavin C | −9.053 |
| Takinib (reference compound) | −8.742 |

C

**Figure 2.** Molecular docking of cannflavins and TAK1 at the ATP binding site in comparison to takinib, an ATP-competitive inhibitor. (**A**) Structural superimposition of the redocked (blue) and experimental native ligand (yellow) at the ATP binding site. (**B**) The free energies of binding of the ligand in complex with the ATP binding site of TAK1. (**C**) The superimposed docking poses of cannflavin A (red), cannflavin B (pink), cannflavin C (green), and takinib (blue).

### 3.3. Molecular Dynamics Simulations and Free Energy Calculations

Validating complex structure stability from docking studies is crucial, as MD simulations provide valuable insights into ligand behavior and intermolecular contacts within the binding site. Based on the free energies of binding and binding pose, the top two docked ligands (cannflavin A and C) and the crystal structure TAK1 with the inhibitor takinib were selected to establish their system stability, flexibility, and other dynamic properties through 200 ns MD simulation using AMBER18.

Root mean square deviation (RMSD) is a method used to assess the dynamic stability of various systems. It quantifies the conformational changes that occur in the protein backbone during the simulation time scale. In this study, the RMSD values were calculated for the TAK1 complex with takinib, cannflavin A, and cannflavin C. As depicted in Figure 3A, all systems reached a state of near stability after 150 ns. The average RMSD values for TAK1–takinib, TAK1–cannflavin A, and TAK1–cannflavin C were determined to be 2.103 Å, 1.945 Å, and 2.068 Å, respectively. Notably, both cannflavin A and C exhibited lower RMSD values compared to the ligand inhibitor takinib, which indicates their relative stability. Throughout the 200 ns simulation, all systems demonstrated their stability profiles, with the TAK1–cannflavin A system exhibiting the highest degree of stability. While the RMSD criterion cannot fully determine the stability of the complex, it is imperative to explore alternative methodologies that enable informed decision-making when selecting candidate compounds for drug discovery.

The compactness of the receptor–ligand complexes during molecular dynamic simulations can be effectively characterized by the radius of gyration (Rg). In this study, the Rg values were calculated to measure the distance between the center of mass of the receptor atoms and their terminals within a specified time frame. A compact receptor structure is indicated by minimal variation in the gyration value, while an expanded or unstable struc-

ture exhibits a higher Rg value. The average Rg values for TAK1–takinib, TAK1–cannflavin A, and TAK1–cannflavin C were found to be 20.334 Å, 19.877 Å, and 20.025 Å, respectively. Notably, all the selected ligands maintained a constant Rg throughout the entire 200 ns simulation, highlighting their compactness in comparison to the TAK1-inhibitor takinib (Figure 3B).

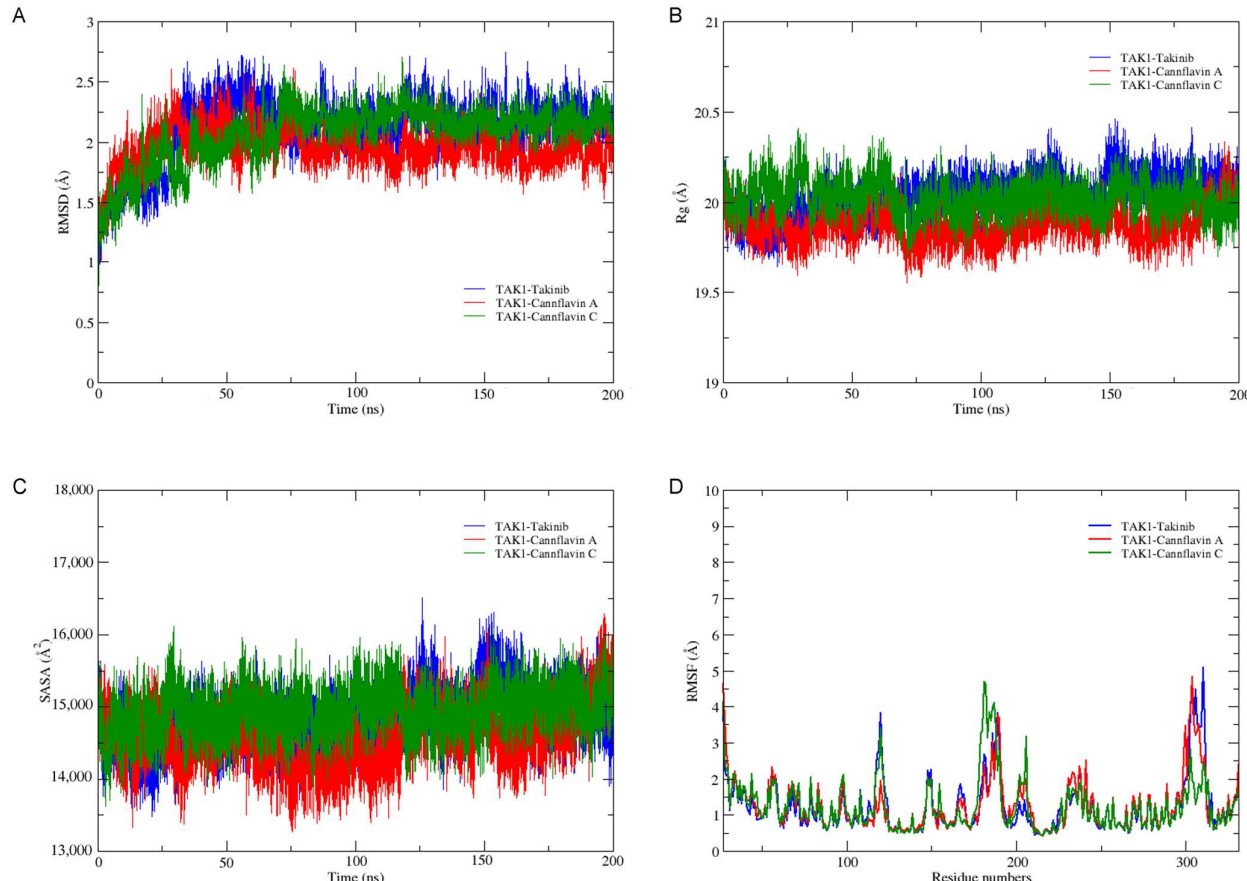

**Figure 3.** The molecular dynamic profiles of stability, compactness, and flexibility in the TAK1 complex systems over a 200 ns simulation. (**A**) RMSD analysis, (**B**) radius of gyration (Rg) analysis, (**C**) SASA analysis, and (**D**) RMSF analysis.

The expansion of protein volume varied across the different systems and was assessed using the solvent-accessible surface area (SASA). SASA describes the portion of a protein that is accessible to a solvent. A higher SASA value indicates an increase in protein volume, and minimal variation over the simulation time is expected. The binding of small molecules (ligands) can modify SASA and potentially influence the protein structure. The average SASA values for TAK1–takinib, TAK1–cannflavin A, and TAK1–cannflavin C were determined to be 14,818.284 Å$^2$, 14,575.268 Å$^2$, and 14,935.170 Å$^2$, respectively. Notably, TAK1–cannflavin C exhibited a higher average SASA value, suggesting system instability. In comparison, TAK1–cannflavin A had lower SASA values than the TAK1–inhibitor takinib, indicating greater stability among the systems (Figure 3C).

The root mean square fluctuation (RMSF) value provides insight into the mobility and flexibility of a protein structure. RMSF analysis was performed on the amino acid residues of the TAK1 complexes with takinib, cannflavin A, and cannflavin C (Figure 3D). It was observed that TAK1–takinib, TAK1–cannflavin A, and TAK1–cannflavin C systems exhibited similar RMSF profiles. However, the low RMSF value of the TAK1–cannflavin A system suggests the formation of stable and rigid complexes between TAK1 and cannflavin A.

Hydrogen bonds are crucial for the formation and stability of the TAK1–ligand complex, as well as for the specific targeting of a receptor by a ligand. In order to identify

the presence of hydrogen bonds, a set of criteria was employed, requiring a distance of 3.0 Å and an angle cut-off of 135°. An analysis of hydrogen bonds was conducted to assess the lifetime and behavior of these interactions between TAK1 and the ligands during a 200 ns molecular dynamics (MD) simulation (see Figure 4). The results revealed that cannflavin A exhibited a greater number of hydrogen bonds compared to the other systems, suggesting that the TAK1–cannflavin A system is the most stable and rigid among all the examined systems.

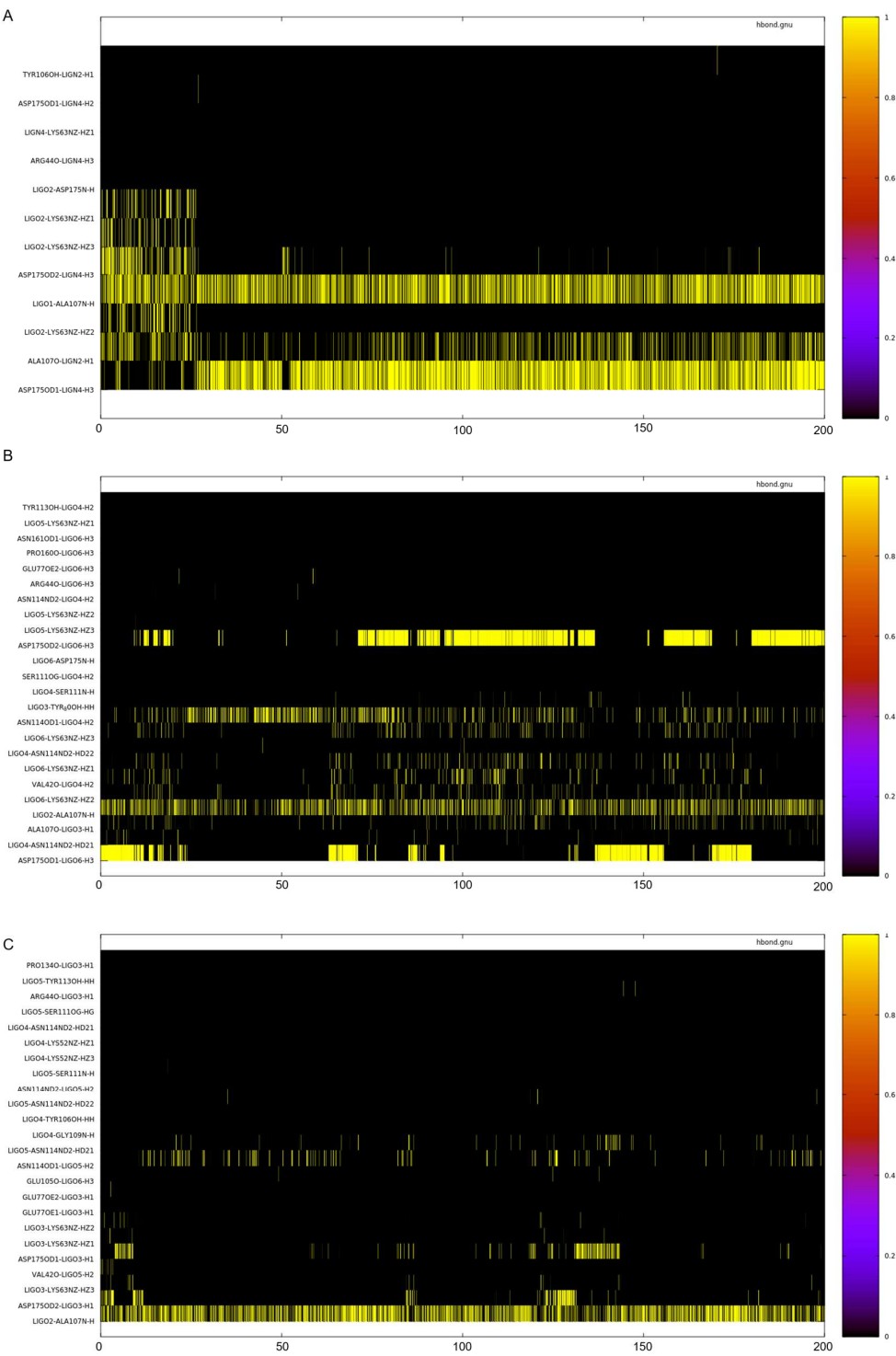

**Figure 4.** Hydrogen bond lifetime profiles of (**A**) takinib, (**B**) cannflavin A, and (**C**) cannflavin C during a 200 ns simulation.

The study of the Gibbs free energy landscape (FEL) provides valuable insights into receptor–ligand complexes. Predicting conformational changes during ligand binding can be achieved through performing FEL analysis on principal component 1 (PC1) and principal component 2 (PC2). FEL allows for the visualization of the global minimum energy conformation of a receptor–ligand complex. In cases where the receptor–ligand interaction is weak or unstable, multiple minimum energy clusters may be observed. Conversely, a strong and stable interaction typically leads to a single conformation cluster in the potential energy distribution. In the FEL plot, the color purple represents lower energy states with highly stable structural conformations, while the red region indicates higher energy conformations. Analysis of Figure 5 reveals that the TAK1–cannflavin A complex exhibits a smaller and more concentrated purple minimal energy area compared to the TAK1–inhibitor takinib complex. This suggests that cannflavin A forms highly stable complexes with TAK1. On the other hand, cannflavin C displays a split basin and fewer purple minimal energy areas, indicating conformational instability in the receptor–ligand complex. These findings suggest that cannflavin A forms structurally favorable complexes with TAK1, characterized by stronger stability and energetics, in comparison to the reference compound takinib.

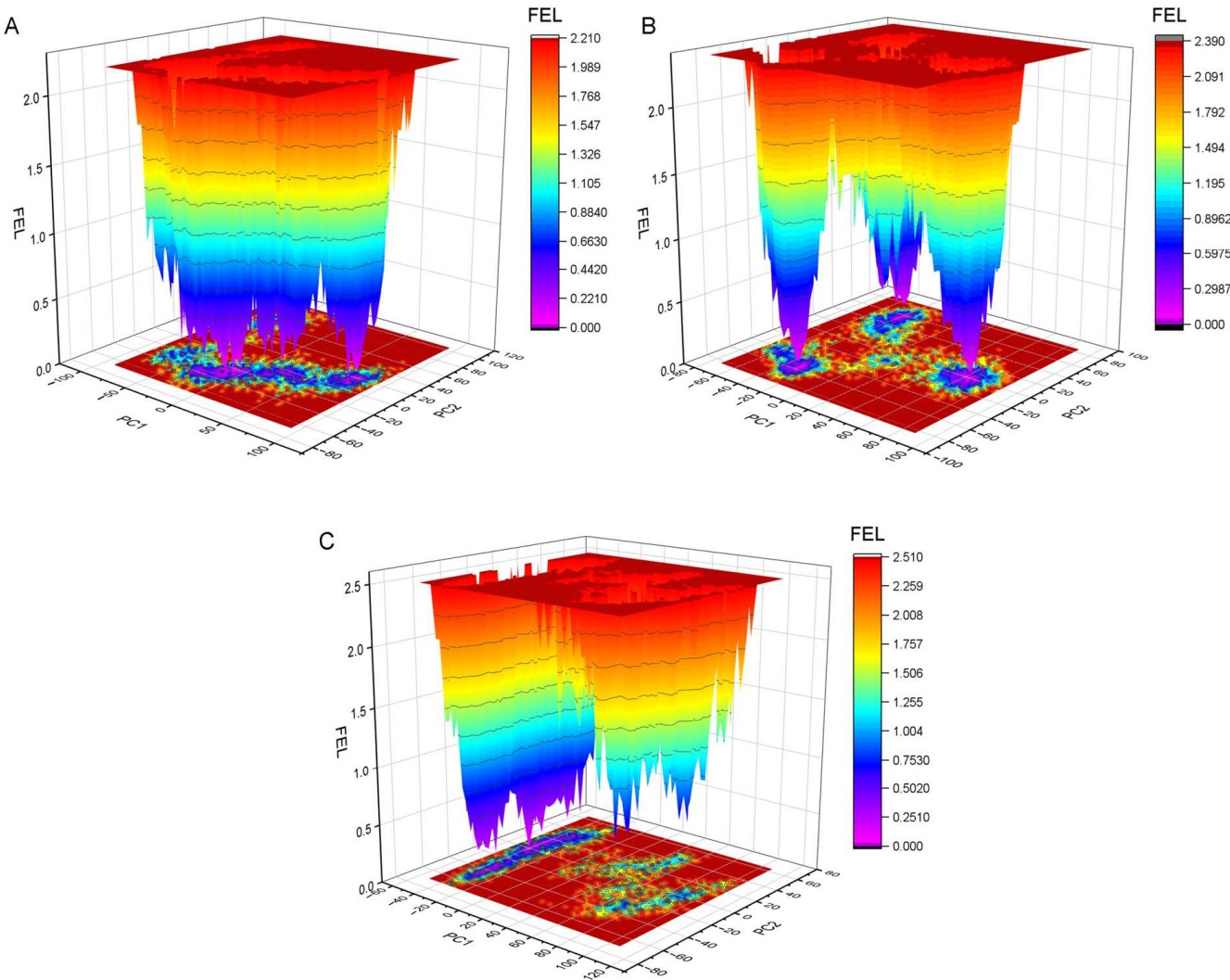

**Figure 5.** The Gibbs free energy landscape (FEL) profiles of (**A**) takinib, (**B**) cannflavin A, and (**C**) cannflavin C during a 200 ns simulation.

Estimating the interaction energy between a protein and ligand through MD trajectory analysis yields valuable insights into ligand affinity and enhances our understanding of molecular interactions. In order to determine a more accurate estimation of the binding free energy between TAK1 and the selected ligands at the receptor inhibition site, the MM/GBSA-based method was employed. This method has been widely utilized for its effectiveness in binding energy calculations. The binding free energy was computed using 1000 snapshots extracted from the last 50 ns of the trajectories. A lower binding free energy value indicates a stronger binding affinity between the receptor and ligand, suggesting a more favorable interaction. As presented in Table 1, cannflavin A exhibited a higher binding affinity (−42.9987 kcal/mol) towards the ATP binding site of TAK1 compared to the inhibitor takinib (−32.5310 kcal/mol). This increased binding affinity can be attributed to the dominant contributions of van der Waals forces and electrostatic energy. The hydrophobic contacts and electrostatic interactions play a crucial role in establishing an efficient binding relationship between the ligands and the binding site of TAK1. These factors are key determinants in the stability and effectiveness of ligands binding to TAK1.

**Table 1.** The binding free energy for the TAK1 complexes using the MM/GBSA method (kcal/mol).

| Compounds | VDW (kcal/mol) | ELE (kcal/mol) | EGB (kcal/mol) | ESURF (kcal/mol) | $\Delta G_{gas}$ (kcal/mol) | $\Delta G_{solv}$ (kcal/mol) | $\Delta G_{TOTAL}$ (kcal/mol) |
|---|---|---|---|---|---|---|---|
| Cannflavin A | −49.3581 | −22.9419 | 36.3489 | −7.0476 | −72.3000 | 29.3013 | −42.9987 |
| Cannflavin C | −49.2620 | −14.3163 | 32.3529 | −6.5341 | −63.5783 | 25.8188 | −37.7595 |
| Takinib | −42.7576 | −18.2630 | 33.9200 | −5.4303 | −61.0206 | 28.4897 | −32.5310 |

VDW, the van der Waals forces; ELE, the electrostatic energy; EGB, the solvation free energy; ESURF, the nonpolar contribution to the solvation free energy; $\Delta G_{gas}$, the gas-phase free energy; $\Delta G_{solv}$, the solvation free energy; $\Delta G_{TOTAL}$, the total free energy.

Per-residue binding energy decomposition was conducted to gain detailed insights into the complexes between TAK1 and its ligands, aiming to identify the key residues responsible for ligand binding (Figure 6). Cannflavin A demonstrated a higher binding affinity, forming three hydrogen bonds with LYS63, ALA107, and ASP175, in addition to engaging in hydrophobic interactions with VAL42, VAL50, ALA61, MET104, TYR106, GLU108, GLY109, GLY110, SER111, ASN114, ALA119, PRO121, LEU163, and CYS175 (Figure 6A). On the other hand, cannflavin C formed two hydrogen bonds with LYS63 and ALA107, along with hydrophobic interactions involving VAL42, GLY43, ARG44, VAL50, ALA61, MET104, TYR106, GLY110, SER111, TYR113, ASN114, PRO160, LEU163, CYS175, and ASP175 (Figure 6B). As for takinib, it established two hydrogen bonds with ALA107 and ASP175 and engaged in hydrophobic interactions with VAL42, GLY43, VAL50, ALA61, VAL90, MET104, GLU105, TYR106, GLY110, SER111, LEU163, and CYS174 (Figure 6C). Pharmacophore modeling studies have revealed that hydrogen bonds formed with LYS63 and ASP175 are critical residues within the ATP binding site of TAK1, contributing to high potency [55]. Furthermore, certain TAK1 inhibitors currently undergoing clinical trials were found to form hydrogen bonds with GLU105 and ALA107 in the hinge region [12]. This finding strongly supports the notion that cannflavin A exhibits a higher binding affinity to TAK1 by means of interacting with both LYS63 (−2.25 kcal/mol) and ASP175 (−7.17 kcal/mol), as well as forming more hydrogen bonds compared to other compounds. The results from the MD simulations and free energy calculations confirmed that cannflavin A significantly inhibited TAK1 via an ATP binding site. This suggests that cannflavin A has the potential to effectively inhibit TAK1 activity, potentially achieving comparable or superior outcomes to takinib. As a result, considering the positive effects of takinib in treating rheumatoid arthritis and the promising targeting properties of cannflavin A towards TAK1, cannflavin A could prove particularly useful in the treatment of rheumatoid arthritis. However, it is essential to emphasize that further preclinical and clinical studies are necessary to validate the efficacy and safety of cannflavin A as a treatment for rheumatoid arthritis.

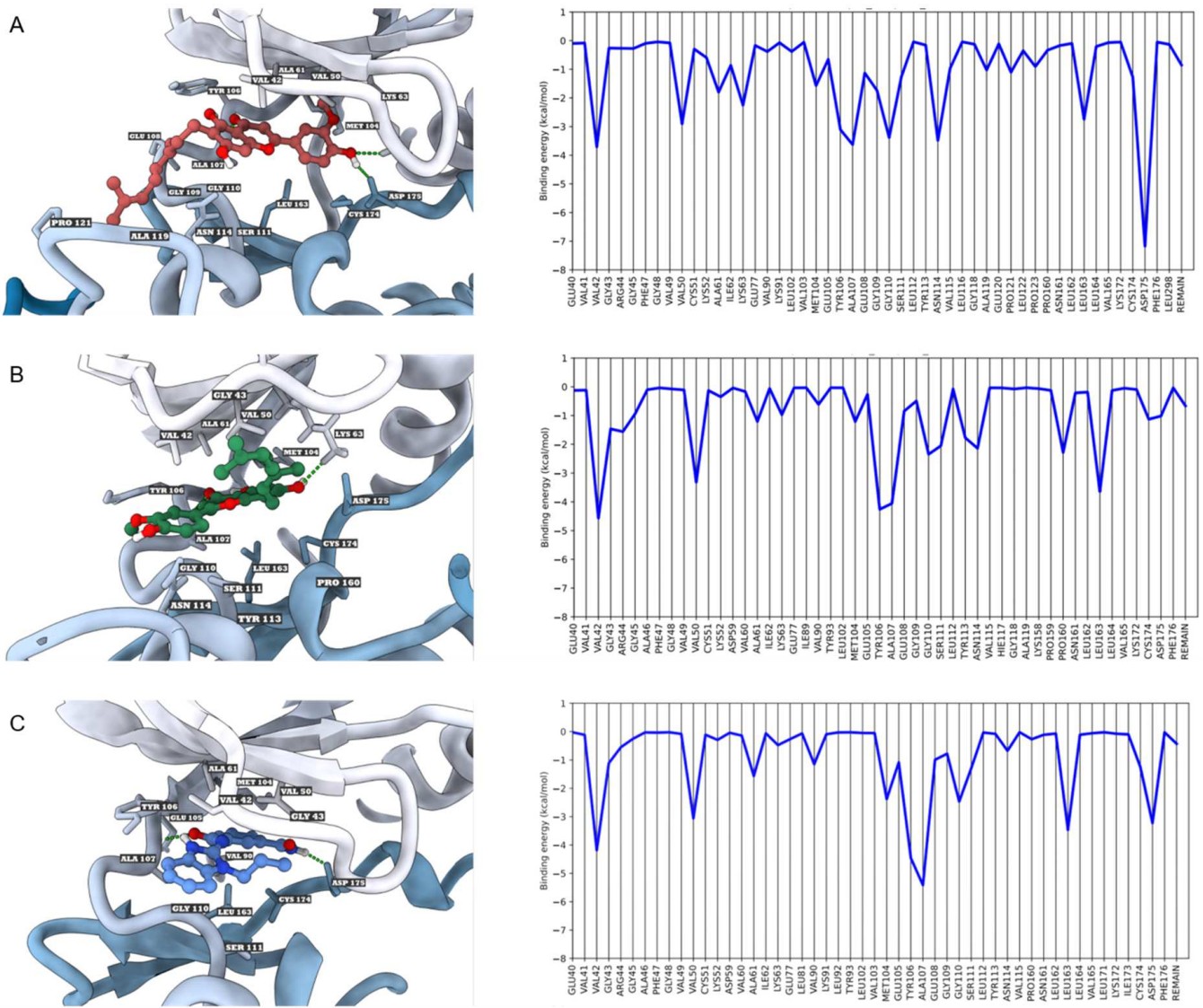

**Figure 6.** The binding interaction and energies of amino acids surrounding the ATP binding site of TAK1: (**A**) cannflavin A, (**B**) cannflavin C, and (**C**) takinib. The green dashed lines denote hydrogen-bonding interaction.

## 4. Conclusions

The Cannabis genus of plant has been recognized as a valuable source of investigational medicinal products. This study investigated the potential of cannflavins derived from *Cannabis sativa* as inhibitors of TAK1. The findings highlight cannflavin A as a promising candidate for TAK1 inhibition, as it exhibits favorable pharmacokinetic and toxicological properties, along with potent anti-inflammatory activity at the atomic level. Further experimental and in vivo studies are needed for validation and assessment of its efficacy and safety as a therapeutic agent for inflammation-related diseases like rheumatoid arthritis.

**Author Contributions:** Conceptualization—K.R. and S.C.; methodology—S.C., S.R. and K.R.; formal analysis—S.C., S.R., B.S., P.P. and K.R.; investigation—S.C., S.R., B.S., P.P. and K.R.; writing and original draft preparation—S.C., S.R. and P.P.; writing, review, and editing—K.R.; supervision, K.R. All authors have read and agreed to the published version of the manuscript.

**Funding:** This research received no external funding.

**Institutional Review Board Statement:** Not applicable.

**Informed Consent Statement:** Not applicable.

**Data Availability Statement:** The datasets used and/or analyzed during the current study are available from the corresponding author upon reasonable request.

**Conflicts of Interest:** The authors declare no conflict of interest.

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
