# Peer review of "Computational Assessment of Cannflavin A as a TAK1 Inhibitor: Implication as a Potential Therapeutic Target for Anti-Inflammation"

_scipharm, doi:10.3390/scipharm91030036_

Round 1

Reviewer 1 Report

In this manuscript the authors investigated the inhibitory potential of Cannabis sativa flavonoids on TAK1 enzyme, and compared the results to the inhibitor takinib through different software simulations. They concluded that the compound cannflavin A is not only a promising inhibitor candidate, but also can have favorable pharmacokinetic and toxicological properties.  The in silico study is properly carried out and can be a solid base for further in vitro investigations. The manuscript itself is well written and structured, I recommend its acception in the present form.

Reviewer 2 Report

The authors reported that cannfla-vin A exhibits excellent ADMET properties and displays superior binding affinity and stability at the ATP binding site of TAK1 when compared to the known inhibitor takinib by in silico approaches. The decomposition of binding free energy unveils critical amino acid residues involved in TAK1 binding in this study are clear. 

Reviewer 3 Report

In the current article the authors used in silico approaches to investigate  the structural insights concerning  the molecular complexation between cannflavin A and TAK1 using ADMET screening, molecular docking, molecular dynamics simulation, free energy calculations and analyzed their drug-likeness properties and toxicity. The results highlight the potential of cannflavin A as a TAK1 inhibitor and its significant implications for the development of targeted therapies in inflammation-related diseases, particularly in rheumatoid arthritis.

Some comments:

-pg 2: you wrote: “Moreover, these agents cannot cross the blood–brain barrier (BBB) due to their high molecular weight, ren-dering them ineffective in treating central nervous system (CNS) diseases associated with TNF signaling, including neurodegenerative conditions”. Which is the connection between polyrheumatoid arthritis and neurodegenerative diseases?

-pg 3: please give details concerning the statement: “These cannflavins were specifically chosen based on compounds found in the previous literature”

-please explain why you consider that cannflavin A could be used particularly in rheumatoid arthritis.

- you wrote that the effect of cannflavins on TAK1 and its underlying molecular mechanism of action has not been demonstrated. I understood that the mechanism of action in unknown, but you didn’t find  in vitro/vivo studies regarding the effect of cannflavins on TAK1? If there are, it would better to introduce them at discussions.

It’s true that nowadays computational analysis have advanced considerably and gained an enormous amount of interest among the researchers. Are useful and valuable tools for understanding the properties behind experimental results but they are far from testing a biological activity itself. The study is interesting and well organized, but, in my opinion, it is much easier to carry out a computational study than an experimental one. So, studies must be continued with in vitro and in vivo determinations.

Minor editing of English language is required.
